# NormFit: A Lightweight Solution for Few-Shot Federated Learning with Non-IID Data

**Azadeh Motamedi**[*]
Queen's University
Kingston, Canada
A.Motamedi@queensu.ca

**Jae–Mo Kang**
Kyungpook National University
Daegu, South Korea
jmkang@knu.ac.kr

**Il–Min Kim**
Queen's University
Kingston, Canada
ilmin.kim@queensu.ca

## Abstract

Vision–Language Models (VLMs) have recently attracted considerable attention in Federated Learning (FL) due to their strong and robust performance. In particular, few-shot adaptation with pre-trained VLMs like CLIP enhances the performance of downstream tasks. However, existing methods still suffer from substantial communication overhead, high local computational demands, and suboptimal performance under non-IID user data. To simultaneously address all those limitations, we propose NormFit, a lightweight solution that selectively fine-tunes only a very small portion of the model parameters, specifically only the Pre-LayerNorm parameters of the vision encoder within a VLM. Overcoming the existing tradeoff between performance and communication/computation efficiency in few-shot FL, NormFit sets a new benchmark by simultaneously achieving superior accuracy and substantially reduced communication and computational demands. Theoretically, we show that NormFit yields a considerably smaller generalization gap compared to tuning all LayerNorm parameters. Importantly, NormFit can function effectively as a standalone solution or integrate seamlessly with existing few-shot fine-tuning methods to further enhance their performance. Notably, NormFit offers implementation simplicity, achieving these improvements without any algorithmic modifications, changes to the underlying model architecture, or the addition of external parameters.[2]

## 1 Introduction

Federated Learning (FL) has emerged as a promising paradigm for decentralized machine learning, allowing multiple users to collaboratively train a shared model without sharing raw data. This approach preserves user data privacy, making it suitable in sensitive domains such as healthcare and finance. However, FL faces several challenges that limit its scalability and effectiveness. As the model size expands, the communication overhead between the central server and users increases significantly. Additionally, the computational demands on the user side substantially grow with larger models and more sophisticated training algorithms. Furthermore, the inherent nature of non-independent and identically distributed (non-IID) users data continues to adversely impact model performance.

Vision-Language Models (VLMs), such as Contrastive Language–Image Pretraining (CLIP) [Radford et al., 2021], pre-trained on large-scale image–text pairs, have recently attracted significant attention in the FL community for their strong zero-shot performance. Pre-trained VLMs are advantageous

---

[*]Corresponding author

[2]The code is available at https://github.com/AziMtmd/NormFit.

39th Conference on Neural Information Processing Systems (NeurIPS 2025).

| Method | | Trainable params (K) | Comm. cost (KB) | Comp. cost (GFLOPs) |
|---|---|---|---|---|
| Full fine-tuning | | $1.5 \times 10^5$ | $4.1 \times 10^5$ | $8.4 \times 10^5$ |
| Standard VLM few-shot methods adopted in FL | CoOp | $2.5 \times 10^1$ | $6.8 \times 10^1$ | $2.3 \times 10^2$ |
| | CoCoOp | $8.9 \times 10^2$ | $2.4 \times 10^3$ | $2.9 \times 10^5$ |
| | TIP-Adapter-F | $7.9 \times 10^2$ | $2.2 \times 10^3$ | $3.1 \times 10^5$ |
| FL VLM few-shot methods | FedCLIP | $5.3 \times 10^2$ | $1.5 \times 10^3$ | $1.0 \times 10^5$ |
| | FedTPG | $3.2 \times 10^2$ | $8.7 \times 10^2$ | $3.3 \times 10^5$ |
| | FedDAT | $7.5 \times 10^2$ | $2.1 \times 10^3$ | $8.5 \times 10^5$ |
| | FLoRA | $2.5 \times 10^1$ | $6.8 \times 10^1$ | $2.8 \times 10^5$ |
| **NormFit (ours)** | | **1.5** | **4.1** | $\mathbf{8.5 \times 10^1}$ |

(a)

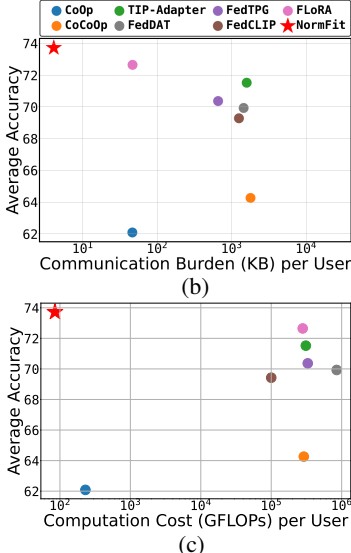

(b)

(c)

Figure 1: (a) Number of trainable parameters (K) per user, communication burden (KB) between the server and each user, and computation cost (GFLOPs) per user in few-shot FL. Averaged accuracy in few-shot FL (over the 10 datasets in Table 1) versus (b) communication burden per user and (c) computation cost per user.

particularly for FL, because the pre-trained knowledge embedded in VLMs can serve as a strong initialization, reducing the number of communication rounds and computation cost required for convergence in FL while providing robust performance on different tasks. Yet, in practice, effectively adapting VLMs to specific downstream tasks typically necessitates few-shot fine-tuning. For VLMs, few-shot learning is commonly achieved either by introducing additional trainable parameters [Zhou et al., 2022b,a, Gao et al., 2024, Zhang et al., 2021, Zanella and Ben Ayed, 2024] or by selectively fine-tuning a subset of existing model parameters [Zaken et al., 2021, Li et al., 2024].

Recently, few-shot fine-tuning for VLMs has also been explored in the context of FL. FedCLIP [Lu et al., 2023] applied adapters to the deeper layers of VLMs, which limits the adaptability of early-layer features—potentially problematic when local data is significantly non-IID. FedTPG [Qiu et al., 2024] adopted personalized prompt tuning for non-IID data across users, aiming to reduce communication overhead and user drift. FedDAT [Chen et al., 2024] fine-tuned small modules inserted into a frozen VLM, adapting models to local distributions while reducing communication costs. FLoRA [Nguyen et al., 2024] introduced small trainable residual adapters in VLMs.

Although existing methods have made meaningful progress in the area, they continue to face two major limitations in non-IID few-shot FL. First, as illustrated in Figure 1(a), these methods still require training a large number of parameters, resulting in substantial communication overhead between users and the server. Particularly, these methods incur high local computation costs, typically involving many local training iterations for convergence, which poses a critical challenge for users with constrained computational resources. This suggests that the parameters selected for training do not converge quickly in non-IID few-shot FL. Second, although these methods have improved performance in few-shot FL, their effectiveness under non-IID conditions remains suboptimal, as illustrated in Figures 1(b) and (c). This unsatisfactory performance is primarily due to those classes with fewer samples, which are inherently more challenging to learn.

To simultaneously address these two challenges, we propose a lightweight few-shot FL solution named *NormFit*, which fine-tunes only the Pre-LayerNorm parameters of the vision encoder within a VLM (in this study, CLIP is adopted as a representative example). NormFit offers two key advantages. First, by training only a minimal number of parameters, it significantly reduces communication overhead between users and the central server, as illustrated in Figure 1(a). Particularly, training of the selected parameters (i.e., the Pre-LayerNorm parameters) converges rapidly in non-IID few-shot FL, resulting in substantially lower local computational costs—a critical benefit for resource-constrained users. Second, fine-tuning the Pre-LayerNorm of the vision encoder achieves state-of-the-art (SOTA) accuracy in non-IID few-shot FL. By adjusting these early-stage parameters, the model effectively adapts to the skewed class distributions that commonly arise in non-IID few-shot settings. As shown in Figures 1(b) and (c), this targeted approach not only achieves superior accuracy but also substantially reduces communication costs and computational burdens.

Two additional major advantages of NormFit are its versatility and implementation simplicity. Norm-Fit can be employed either as a standalone efficient few-shot solution or it can be seamlessly integrated as an enhancement to existing few-shot learning methods. Also, NormFit achieves all the improvements without requiring any algorithmic modifications, changes to the underlying model architecture, or additional external parameters. In this paper, we also provide a theoretical justification supporting the effectiveness of NormFit. Our contributions are summarized as follows:

- *A Novel, Effective, and Efficient Fine-Tuning Strategy:* We propose NormFit, a new fine-tuning method that selectively updates only the Pre-LayerNorm parameters of the vision encoder in a VLM. NormFit overcomes the existing tradeoff between accuracy and efficiency in non-IID few-shot FL, establishing a new benchmark.

- *Theoretical Justification:* We theoretically demonstrate that NormFit leads to a smaller generalization gap, explaining why NormFit performs robustly in few-shot non-IID FL scenarios.

- *Comprehensive Validation of Superior Performance:* Through extensive experiments on diverse datasets, we empirically demonstrate that NormFit achieves SOTA accuracy while significantly reducing communication and computational costs compared to existing methods.

- *Flexible Integration and Versatility:* We empirically show that NormFit can serve either as an effective standalone method or as a complementary add-on that further improves the performance of existing few-shot fine-tuning techniques without requiring additional modifications.

- *Implementation Simplicity:* All improvements introduced by NormFit require no algorithmic modifications, changes to the underlying model architecture, or additional external parameters.

## 2   Related Works

In this section, we review the existing methods designed to address non-IID data in few-shot FL.

### 2.1   Adapter Tuning in Few-Shot FL

CLIP-Adapter [Gao et al., 2021] and Tip-Adapter [Zhang et al., 2021] are two prominent adapter tuning methods in the centralized setting. CLIP-Adapter proposed a lightweight bottleneck architecture to leverage the knowledge stored in CLIP and the newly acquired knowledge from few-shot samples. Instead of the adapter, Tip-Adapter constructed a query-key cache model to derive the adapter's weights, achieving better performance and lower computational complexity compared to CLIP-Adapter. In the FL setting, FedCLIP [Lu et al., 2023] introduced an attention-based adapter for CLIP in FL, enabling efficient personalization and generalization across users. FLoRA [Nguyen et al., 2024], inspired by Low-Rank Adaptation (LoRA) [Zanella and Ben Ayed, 2024], focused on updating a small subset of the model's parameters in FL, specifically by adding trainable low-rank adaptation modules to certain layers of a pre-trained model. While adapter tuning performs well in centralized learning, it often struggles to effectively address the issue of non-IID data in FL.

### 2.2   Prompt Tuning in Few-Shot FL

Prompt tuning typically optimizes task-related textual tokens in order to embed task-specific knowledge for prediction. The hand-crafted template "a photo of a [Class]" in CLIP is used to model the class embedding for zero-shot prediction. Context Optimization (CoOp) [Zhou et al., 2022b] and Conditional Context Optimization (CoCoOp) [Zhou et al., 2022a], the well-known prompt tuning methods in centralized learning, replaced the hand-crafted prompts with a set of learnable prompts inferred by the labeled few-shot samples. pFedPrompt [Guo et al., 2023] and FedPrompt [Zhao et al., 2023] explored prompt tuning in FL, aiming to create personalized models while addressing the issue of non-IID data. FedTPG [Qiu et al., 2024] proposed a task-specific prompt generation method to enhance the relevance of the generated prompts to each user's data, improving the performance on personalized tasks. FedAPT [Su et al., 2024] extends the idea of CoCoOp to FL, emphasizing domain-specific adaptation across users. Despite being promising, prompt tuning is often less effective in adapting to non-IID data as it inherently relies on a degree of consistency in the underlying distributions [Zhou et al., 2022b,a]. In federated prompt tuning, there is a risk that prompts may overfit to a user's data, reducing their ability to generalize well across users. Also, prompt tuning requires frequent updates to parameters leading to increased communication cost.

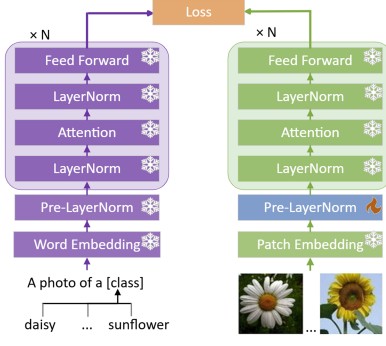

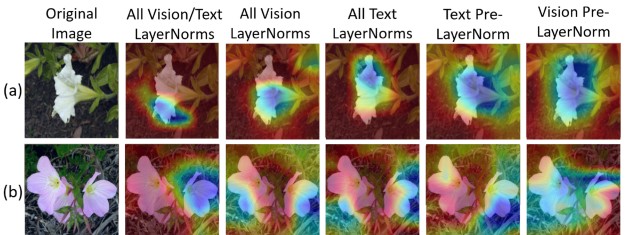

Figure 2: NormFit fine-tunes only the Pre-LayerNorm of CLIP's vision encoder to adapt feature statistics at the input stage. Snowflake and flame icons denote frozen parameters and fine-tuned parameters, respectively.

Figure 3: Grad-CAM visualizations illustrating the effect of different LayerNorm fine-tuning strategies in CLIP. Each column corresponds to a specific tuning configuration: all LayerNorms of both vision and text encoders, all vision LayerNorms, all text LayerNorms, text Pre-LayerNorm only, and vision Pre-LayerNorm only (NormFit). Rows (a) and (b) correspond to the majority and minority classes, respectively. Fine-tuning only the vision Pre-LayerNorm yields sharper and more semantically aligned attention on both classes, notably better focus on the minority class.

## 3 Proposed Method: NormFit

NormFit is a lightweight fine-tuning strategy for VLMs (in this study, CLIP is adopted as a representative example) in few-shot FL. As depicted in Figure 2, its core idea is to adapt only the Pre-LayerNorm parameters of the CLIP vision encoder while freezing all other weights —the remainder of the vision encoder and the entire text encoder. Fine-tuning utilizes a few labeled images per class and textual prompts of the form "a photo of a [class]". During each FL round, users locally perform back-propagation on those minimal parameters (i.e., Pre-LayerNorm) and transmit these updates to the central server. The server aggregates these parameters and broadcasts the aggregated values back to users. Updating the minimal parameters ensures very low communication, while preserving the rich prior knowledge captured during CLIP's pre-training. Moreover, as the Pre-LayerNorm resides at the earliest stage of the model, adapting its parameters facilitates a rapid and effective alignment of the model's feature representations with the local data distribution, decreasing local computational costs. Furthermore, fine-tuning the Pre-LayerNorm enhances the representation quality for underrepresented classes and, consequently, overall accuracy under non-IID conditions.

In the following, we provide detailed discussions explaining why fine-tuning only the Pre-LayerNorm parameters of the vision encoder is effective, followed by mathematical justification to support these discussions. In Section 4, we will confirm the discussions and analysis through extensive experiments.

### 3.1 Which to Fine-Tune: Vision Encoder or Text Encoder?

Our answer to this question is *the vision encoder* because it is more sensitive to distribution shift. Visual inputs often exhibit greater variability than their textual counterparts due to domain-specific differences in lighting, background, scale, etc. This makes the visual features more prone to domain misalignment. Previous work shows that even lightweight adaptations can meaningfully recalibrate the internal statistics of the visual backbone. For example, the authors in [Fahes et al., 2024] demonstrate that fine-tuning the visual projection head outperforms prompt tuning, suggesting that the vision encoder is the primary driver of adaptation quality in few-shot image classification.

By contrast, the text encoder is inherently more robust. CLIP text encoder captures semantically rich, stable features that generalize across domains [Radford et al., 2021]. As noted in CoOp [Zhou et al., 2022b], modifying the text side yields only modest improvements, and performance is relatively insensitive to the capacity of the text encoder. Prompt tuning often leverages this stability by keeping the text encoder frozen and instead learning task-adaptive prompts [Zhou et al., 2022a]. While this avoids catastrophic forgetting, it limits flexibility under severe non-IID scenarios.

These findings suggest a consistent trend: when constrained to fine-tune a small subset of parameters, allocating those to the vision encoder yields more substantial gains than adapting the text encoder. Related work spans multiple settings, including image classification [Fahes et al., 2024, Zhou et al.,

2022b] and federated settings [Lu et al., 2023]. Text encoder tuning can be useful in some settings but often requires vision-side updates to reach peak performance.

## 3.2 Why Fine-Tune Only the Pre-LayerNorm?

There are two primary reasons. First, the Pre-LayerNorm contains a very small number of parameters; updating only these parameters helps preserve the rich prior knowledge embedded in the pretrained model. In addition, because the Pre-LayerNorm is located at the earliest stage of the model, adapting its parameters rapidly aligns the entire model's feature representations with local data distributions. In the context of FL, these minimal and earliest-stage updates significantly reduce communication and computation costs at each user, which is particularly beneficial in resource-constrained environments.

Second, fine-tuning the Pre-LayerNorm is effective for improving the representational quality, especially for the classes with fewer samples (namely, minority classes), which are inherently more challenging to learn. Grad-CAM visualizations in Figure 3 show the impact of various LayerNorm fine-tuning strategies applied to the CLIP model in a FL setting with 10 users. The columns represent different tuning configurations: fine-tuning all LayerNorms of both vision and text encoders, all LayerNorms of vision encoder, all LayerNorms of text encoder, text encoder's Pre-LayerNorm only, and finally, vision encoder's Pre-LayerNorm only, which is NormFit. Row (a) displays an example from a majority class (having an average of 10 samples per user), and row (b) provides an example of a minority class (having an average of 3 samples per user). Fine-tuning only the vision encoder's Pre-LayerNorm (NormFit) generates sharper, more precise attention maps for both majority and minority classes. The benefit of NormFit is particularly notable for the minority class with an improvement in the clarity and relevance of highlighted regions. This suggests that selectively tuning those earliest-stage parameters in the vision encoder can effectively mitigate biases from class imbalance, enhancing model robustness and predictive accuracy in few-shot non-IID scenarios.

## 3.3 Theoretical Justification

We analytically substantiate why NormFit, which only fine-tunes the Pre-LayerNorm, provides excellent generalization performance. Specifically, we analyze the generalization gap, which quantifies the discrepancy between the model's performance on training samples (empirical risk) and its expected performance on unseen data (true risk). A smaller generalization gap signifies that the model has captured meaningful patterns rather than simply memorizing training instances, thereby enhancing its robustness in diverse scenarios. We formally define this concept and rigorously analyze it within the FL framework introduced in [Mohri et al., 2019] in the following.

**Definition 1 (Generalization Gap $\varepsilon$ in FL) [Mohri et al., 2019]:**

*Let $\ell(h(x), y)$ denote a loss function evaluated on an input-output pair $(x, y)$. In FL with $M$ users, if each user $k$ has its own distribution $\mathcal{D}_k$, then for any weight vector $\vec{\lambda} = (\lambda_1, \ldots, \lambda_M)$ in the probability simplex, the target distribution is defined as $\mathcal{D}_{\vec{\lambda}} = \sum_{k=1}^{M} \lambda_k \mathcal{D}_k$ and the true risk of a hypothesis $h$ is given by $L_{\mathcal{D}_{\vec{\lambda}}}(h) = \mathbb{E}_{(x,y) \sim \mathcal{D}_{\vec{\lambda}}}[\ell(h(x), y)]$. Similarly, if each user $k$ has its own empirical distribution (based on its local data), denoted by $\hat{\mathcal{D}}_k$, the empirical risk on the mixture is $L_{\hat{\mathcal{D}}_{\vec{\lambda}}}(h) = \mathbb{E}_{(x,y) \sim \hat{\mathcal{D}}_{\vec{\lambda}}}[\ell(h(x), y)]$, where $\hat{\mathcal{D}}_{\vec{\lambda}} = \sum_{k=1}^{M} \lambda_k \hat{\mathcal{D}}_k$. The generalization gap $\varepsilon$ is defined as:*

$$\varepsilon(h) := \sup_{\vec{\lambda} \in \Lambda} \left[ L_{\mathcal{D}_{\vec{\lambda}}}(h) - L_{\hat{\mathcal{D}}_{\vec{\lambda}}}(h) \right]. \tag{1}$$

In the following theorems, we present a foundational analysis comparing the impact of fine-tuning only the Pre-LayerNorm versus fine-tuning all LayerNorms on the model's generalization performance.

**Theorem 1 (Generalization Gap Ratio $\varepsilon_{all}/\varepsilon_{pre}$ in FL: Pre-LayerNorm vs. All LayerNorms Fine-Tuning within the Vision Encoder):**

*Let $\varepsilon_{pre}$ and $\varepsilon_{all}$ denote the generalization gaps under the same FL setup as in [Mohri et al., 2019] when fine-tuning (i) only the Pre-LayerNorm and (ii) all LayerNorms, respectively, for a Vision Transformer (ViT) with a Pre-LayerNorm and $N$ transformer layers each containing two LayerNorms*

*with embedding dimension d. Then the ratio of the generalization gaps is bounded as follows:*

$$\frac{\varepsilon_{all}}{\varepsilon_{pre}} \lesssim (2N+1)\sqrt{\left(1 + \frac{\log(2N+1)}{\log(2d)}\right)}. \tag{2}$$

**Proof.** The proof is provided in Appendix A. ∎

If we consider ViT-B/32 (base vision encoder model in the standard CLIP), we have $N = 12$ and $d = 768$. Then the bound becomes:

$$\frac{\varepsilon_{all}}{\varepsilon_{pre}} \lesssim 29.98. \tag{3}$$

This suggests that, for the standard CLIP, the generalization gap when fine-tuning all LayerNorms could be up to about 30 times larger than when fine-tuning a single LayerNorm, all else being equal. This large multiplicative factor provides a clear theoretical rationale for why NormFit achieves robust performance with a minimal parameter update.

Complementing Theorem 1, we present Theorem 2, which analytically demonstrates that the optimal choice is the Pre-LayerNorm, rather than any deeper alternative in the network. This result highlights the unique influence of the Pre-LayerNorm within the network. To formalize this locational advantage, we leverage Lipschitz constants, which are an essential tool in deep neural network analysis.

**Definition 2 (Lipschitz Continuous Function) [Boyd and Vandenberghe, 2004]:** *A function $f : \mathbb{R}^n \to \mathbb{R}^m$ is defined as Lipschitz continuous if there exists a non-negative constant $L$ such that for any two inputs $x_1, x_2$, the following inequality holds:*

$$\|f(x_1) - f(x_2)\| \leq L\|x_1 - x_2\|. \tag{4}$$

*The smallest such $L$ is the Lipschitz constant of $f$.*

In the following theorem, we provide the gradient of the global Lipschitz bound with respect to the parameters of any arbitrary LayerNorm (including the Pre-LayerNorm as a special case).

**Theorem 2 (Influence of Fine-tuning a LayerNorm):**

*Let $f_\Theta$ be a ViT of $N$ layers, $f = f_N \circ \cdots \circ f_1$, where $\Theta$ represents the set of trainable parameters of all LayerNorms in the ViT. All other parts, including attention blocks and MLPs are frozen. Let $L_{global}(\Theta) = \prod_{l=1}^{N} L_l(\Theta)$ be an upper bound on its global Lipschitz constant, where $L_k(\Theta)$ is the Lipschitz constant of layer $k$. Let $\Theta_k$ denote the set of trainable parameters of the $k$-th LayerNorm (in layer $k$). The gradient of the global Lipschitz bound with respect to $\Theta_k$ is given by:*

$$\nabla_{\Theta_k} L_{global} = DIT_k + CIT_k, \tag{5}$$

*where the Direct Influence Term, $DIT_k$, is given by*

$$DIT_k = (\nabla_{\Theta_k} L_k) A_k, \tag{6}$$

*with an amplification factor*

$$A_k = \left(\prod_{i=1}^{k-1} L_i\right)\left(\prod_{i=k+1}^{N} L_i\right), \tag{7}$$

*and the Cascading Influence Term, $CIT_k$, is given by*

$$CIT_k = \sum_{j=k+1}^{N} (\nabla_{\Theta_k} L_j)\left(\prod_{i \neq j} L_i\right). \tag{8}$$

**Proof.** The proof is provided in Appendix B. ∎

Theorem 2 reveals the unique locational influence of Pre-LayerNorm. First, in the Direct Influence Term, we have $A_1 > A_k$, which means that parameters of the first LayerNorm possess a uniquely amplified influence on the global Lipschitz bound, positioning them at a point of maximum leverage

for model regularization. To see why $A_1 > A_k$ holds, note that the amplification factor for $k = 1$ is $A_1 = \prod_{j=2}^{N} L_j$, while for $k > 1$ it becomes $A_k = \prod_{j \neq k} L_j = (L_1/L_k)A_1$. Clearly, $A_1 > A_k$ holds if and only if $L_1 < L_k$. Pre-LayerNorm at $k = 1$ receives input directly from the patch embeddings, which have a well-controlled, low-variance distribution. In contrast, a LayerNorm at any deeper layer $k > 1$ receives hidden states that have been progressively amplified by the preceding $k - 1$ transformer blocks. These hidden states exhibit higher variance due to the accumulation of nonlinear transformations and residual connections. To handle this increased variance, deeper layers must be more expansive, leading to larger effective Lipschitz constants; that is, $L_1 < L_k$.

Second, $\text{CIT}_k$ is also different. For the Pre-LayeNorm ($k = 1$), $\text{CIT}_k$ is a sum of terms describing how changes in $\Theta_1$ propagate to affect the Lipschitz constants of all $(N - 1)$ subsequent layers, $2, \ldots, N$. In contrast, for any deeper layer $k > 1$, $\text{CIT}_k$ is a sum of terms describing how changes in $\Theta_k$ propagate to affect the Lipschitz constants of only the $(N - k)$ subsequent layers, $k + 1, \ldots, N$.

Intuitively, changes to the Pre-LayerNorm parameters ($k = 1$) have a direct impact that is amplified by the entire depth of the network, alongside a cascading effect that ripples through every subsequent layer. This provides a powerful, global control. In contrast, changes to deeper LayerNorm parameters ($k > 1$) have a direct impact that is amplified by a structurally different, and smaller, product of constants. Also, the cascading effect is more localized, influencing only the remaining deeper layers.

## 4    Experiments

### 4.1    Simulation Setting

**Datasets:** We present our results on a variety of datasets, including CIFAR10 [Krizhevsky et al., 2009], ImageNet [Deng et al., 2009], Caltech101 [Fei-Fei et al., 2004], OxfordPet [Parkhi et al., 2012], Cars [Krause et al., 2013], Flowers102 [Nilsback and Zisserman, 2008], Food101 [Bossard et al., 2014], and FGVCAircraft [Maji et al., 2013] for classification; SUN397 [Xiao et al., 2010] for scene recognition; and DTD [Cimpoi et al., 2014] for texture classification.

**Baselines:** As the most representative VLM few-shot methods, we adopted CoOp [Zhou et al., 2022b] and CoCoOp [Zhou et al., 2022a] as baselines for prompt tuning; and TIP-Adapter [Zhang et al., 2021] as a baseline for adapter tuning in FL (Table 1). Additionally, we compare our approach against SOTA few-shot FL methods for VLMs, including FedTPG [Qiu et al., 2024] as a baseline for prompt tuning; and FedCLIP [Lu et al., 2023], FedDAT [Chen et al., 2024], and FLoRA [Nguyen et al., 2024] as baselines for adapter tuning (e.g., in Table 2).

**Evaluation metrics:** Each user evaluates the model using the global test set, as our primary goal is to develop a global model in FL rather than a personalized one. The evaluation emphasizes the task-specific performance metric, top-1 accuracy. Furthermore, we assess the communication efficiency (between the server and each user) and the computation cost (per user).

**Implementation details:** We consider the FL setting of 100 users, each having a CLIP ViT-B/32 model. The learning rate, batch size, and weight decay are $5 \times 10^{-4}$, 64, and $10^{-4}$, respectively. All experiments were conducted on a single A100 GPU with 48 GB of RAM. To create non-IID data, the widely-adopted Dirichlet distribution [Yurochkin et al., 2019] with a small value of concentration parameter $\beta = 0.1$ is used and the maximum number of samples per class is limited to 16 (unless otherwise specified). More details for the non-IID setting and the data distribution visualizations are provided in Figures A.1 and A.2 in Appendix D.

### 4.2    Simulation Results: NormFit as a Standalone Strategy

NormFit's flexibility allows it to function as a standalone fine-tuning strategy or as an add-on to existing fine-tuning methods. We evaluate NormFit as a standalone solution in this subsection and, in Section 4.3, we assess its performance as an add-on integrated with existing fine-tuning techniques.

#### 4.2.1    Comparison to the Most Representative VLM Fine-Tuning Methods Adopted in FL

Table 1 shows the accuracy of NormFit and the most representative VLM few-shot fine-tuning methods adopted in FL in non-IID scenarios with $\beta = 0.1$. We report the results for a more skewed non-IID setting with $\beta = 0.05$ in Table A.2 in Appendix D. All results demonstrate that

Table 1: Accuracy of NormFit and the most representative VLM few-shot methods adopted in FL.

| Baseline | CIFAR10 | ImageNet | Caltech | OxfordPets | Cars | Flowers | Food | Aircraft | SUN | DTD | Average |
|---|---|---|---|---|---|---|---|---|---|---|---|
| CoOp | 78.71±0.19 | 50.33±0.18 | 83.81±0.14 | 74.48±0.17 | 51.90±0.17 | 82.60±0.13 | 64.90±0.15 | 20.36±0.15 | 58.42±0.11 | 55.32±0.16 | 62.08±0.16 |
| CoCoOp | 81.90±0.16 | 51.72±0.16 | 88.36±0.10 | 86.59±0.11 | 60.30±0.12 | 73.17±0.12 | 73.09±0.10 | 20.43±0.11 | 60.96±0.13 | 50.53±0.13 | 64.71±0.12 |
| TIP-Adapter | 89.14±0.71 | 62.14±0.81 | 88.11±0.94 | 85.27±0.90 | 70.12±0.80 | 89.01±0.66 | 72.61±0.60 | 30.61±0.32 | 63.84±0.93 | 57.25±0.50 | 70.81±0.72 |
| NormFit (ours) | **89.54±0.07** | **62.57±0.21** | **89.09±0.06** | **90.22±0.18** | **70.46±0.25** | **91.27±0.08** | **81.06±0.31** | **32.59±0.44** | **70.01±0.62** | **60.42±0.10** | **73.72±0.23** |

Table 2: Accuracy of NormFit and the most recent FL VLM few-shot methods.

| Baseline | CIFAR10 | ImageNet | Caltech | OxfordPets | Cars | Flowers | Food | Aircraft | SUN | DTD | Average |
|---|---|---|---|---|---|---|---|---|---|---|---|
| FedCLIP | 85.96±0.10 | 58.46±0.13 | 85.79±0.14 | 86.96±0.14 | 68.20±0.17 | 87.12±0.16 | 76.45±0.12 | 21.55±0.04 | 64.21±0.08 | 58.55±0.02 | 69.33±0.11 |
| FedTPG | 86.44±0.22 | 59.92±0.17 | 86.74±0.16 | 88.23±0.19 | 68.13±0.18 | 87.46±0.20 | 78.57±0.11 | 22.46±0.19 | 66.39±0.27 | 59.32±0.04 | 70.37±0.17 |
| FedDAT | 84.69±0.34 | 61.70±0.40 | 83.43±0.29 | 87.57±0.36 | 66.99±0.19 | 88.69±0.42 | 78.25±0.47 | 23.68±0.20 | 65.23±0.21 | 60.88±0.23 | 70.11±0.31 |
| FLoRA | 88.49±0.36 | 61.29±0.26 | 87.93±0.27 | 90.02±0.15 | 70.17±0.23 | 90.76±0.09 | 79.35±0.16 | 32.22±0.25 | 67.16±0.12 | 59.09±0.14 | 72.64±0.21 |
| NormFit (ours) | **89.54±0.07** | **62.57±0.21** | **89.09±0.06** | **90.22±0.18** | **70.46±0.25** | **91.27±0.08** | **81.06±0.31** | **32.59±0.44** | **70.01±0.62** | **60.42±0.10** | **73.72±0.23** |

NormFit exhibits robust performance, mostly outperforming all baseline methods adopted in FL, while substantially reducing communication burden and computation cost, as shown in Figure 1(a).

Prompt tuning methods like CoOp and CoCoOp fail to effectively steer the model toward the underrepresented classes due to two reasons. First, prompts cannot modify the frozen backbone, which limits their ability to represent rare or minority classes. Second, prompts may overfit to the majority classes in few-shot scenarios. Adapter tuning, although not entirely immune to the class imbalance, provides better robustness than prompt tuning, while still under-performing NormFit.

Figure 4 shows Grad-CAM visualizations comparing NormFit with CoOp and CLIP-Adapter for one majority and one minority class. Both CoOp and CLIP-Adapter experience performance degradation for the minority class, indicating difficulties in capturing meaningful representations of underrepresented classes. In contrast, NormFit consistently produces clear and meaningful activation patterns for the minority class, demonstrating its superior capability in managing class-imbalanced data.

### 4.2.2 Comparison to Most Recent FL Few-Shot Methods for VLMs

Table 2 compares NormFit to the most recent FL fine-tuning methods designed for VLMs, including FedCLIP [Lu et al., 2023], FedTPG [Qiu et al., 2024], FedDAT [Chen et al., 2024], and FLoRA [Nguyen et al., 2024]. FedCLIP trains an adapter federally, yet it does not modify CLIP's core parameters, leaving representations biased toward the pretraining data and limiting generalization. FedTPG learns a unified prompt generation network but lacks the capacity to fully address non-IID data. FedDAT uses data augmentation to handle non-IID data, but it risks overfitting by generating slight variations of minority samples rather than new data. FLoRA integrates LoRA adapters into CLIP's text encoder, reducing communication cost and improving adaptability. NormFit outperforms all those methods, while further reducing communication overhead and computation cost (Figure 1).

### 4.3 Simulation Results: NormFit as an Add-On

We now evaluate the effectiveness of NormFit as an add-on method. Table 3 presents the simulation results combining NormFit with CoOp and TIP-Adapter in non-IID few-shot FL scenarios. The results clearly demonstrate that NormFit seamlessly enhances existing few-shot fine-tuning methods for VLMs, significantly improving their performance in non-IID few-shot FL.

### 4.4 Communication Burden and Computation Cost

We assess the communication burden of NormFit against other baselines by examining the size of per-user updates (in KB) shown in Figure 1(a). Because NormFit only transmits the Pre-LayerNorm parameters of the vision encoder (approximately 1.5 K parameters per user), the total data exchanged between the server and each user is only 4.1 KB. In contrast, other methods send tens to hundreds of times more parameters. This substantial reduction makes NormFit especially attractive for environments where bandwidth is scarce.

In Figure 1(a), the computation cost savings of NormFit are even more pronounced, which arise from two key design choices. First, whereas baseline methods add and train new external modules that require random initialization, NormFit updates only existing LayerNorm parameters, leveraging their

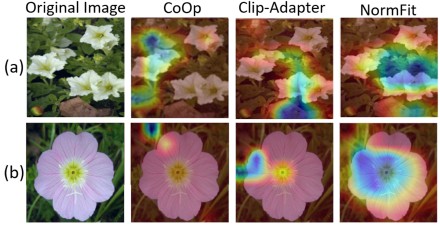

Figure 4: Grad-CAM visualizations for the Flowers dataset, comparing CoOp, CLIP-Adapter, and Norm-Fit. Rows (a) and (b) correspond to the majority and minority classes, respectively. NormFit yields the best semantically aligned attention on both classes, notably better focus on the minority class.

Table 3: Accuracy of NormFit as an add-on to existing fine-tuning methods in few-shot FL.

| Baseline | CIFAR10 | ImageNet | Caltech | OxfordPets | Cars | Flowers | Food | Aircraft | SUN | DTD | Average |
|---|---|---|---|---|---|---|---|---|---|---|---|
| CoOp | 78.71±0.19 | 50.33±0.18 | 83.81±0.14 | 74.48±0.17 | 51.90±0.17 | 82.60±0.13 | 64.90±0.15 | 20.36±0.15 | 58.42±0.11 | 55.32±0.16 | 62.08±0.16 |
| CoOp + NormFit | **88.23**±0.33 | **60.76**±0.24 | **89.17**±0.18 | **91.44**±0.26 | **72.73**±0.31 | **91.94**±0.20 | **81.45**±0.18 | **33.05**±0.24 | **69.84**±0.30 | **60.38**±0.29 | **73.90**±0.25 |
| TIP-Adapter | 89.14±0.71 | 62.14±0.81 | 88.11±0.94 | 85.27±0.90 | 70.12±0.80 | 89.01±0.66 | 72.61±0.60 | 30.61±0.32 | 63.84±0.93 | 57.25±0.50 | 70.81±0.72 |
| TIP-Adapter + NormFit | **91.26**±0.92 | **62.84**±0.97 | **90.23**±1.03 | **88.17**±1.07 | **72.94**±0.94 | **91.65**±0.89 | **75.43**±0.84 | **34.70**±0.40 | **66.17**±1.14 | **59.26**±0.63 | **73.27**±0.88 |

Table 4: NormFit vs. fine-tuning additional LayerNorms in few-shot FL.

| Method | CIFAR10 | ImageNet | Caltech | OxfordPets | Cars | Flowers | Food | Aircraft | SUN | DTD | Average |
|---|---|---|---|---|---|---|---|---|---|---|---|
| All LayerNorms | 88.53±0.13 | 61.36±0.25 | 86.92±0.17 | 84.24±0.10 | 63.92±0.10 | 86.47±0.16 | 78.80±0.20 | 26.20±0.28 | 66.05±0.08 | 55.88±0.14 | 69.84±0.16 |
| Vision encoder LayerNorms | **90.15**±0.09 | 62.45±0.08 | 88.62±0.22 | 87.05±0.24 | 69.14±0.10 | 89.78±0.07 | 80.11±0.23 | 32.12±0.16 | 69.09±0.18 | 58.70±0.24 | 72.72±0.16 |
| Text encoder LayerNorms | 88.34±0.19 | 61.28±0.06 | 87.16±0.08 | 86.40±0.16 | 67.27±0.17 | 89.06±0.09 | 79.53±0.12 | 30.66±0.15 | 67.14±0.11 | 56.02±0.28 | 71.29±0.14 |
| Text encoder Pre-LayerNorm | 89.32±0.17 | 60.93±0.07 | 88.23±0.07 | 87.30±0.10 | 69.84±0.11 | 88.25±0.10 | 80.06±0.13 | 32.47±0.10 | 69.16±0.14 | 58.24±0.10 | 72.38±0.11 |
| NormFit (ours) | 89.54±0.07 | **62.57**±0.21 | **89.09**±0.06 | **90.22**±0.18 | **70.46**±0.25 | **91.27**±0.08 | **81.06**±0.31 | **32.59**±0.44 | **70.01**±0.62 | **60.42**±0.10 | **73.72**±0.23 |

pretrained values. Second, by restricting updates to the Pre-LayerNorm at the network's input stage, NormFit rapidly realigns feature statistics using minimal additional FLOPs. Consequently, each user incurs only $8.5 \times 10^1$ GFLOPs—substantially lower than those competing methods.

## 4.5 Ablation Study

### 4.5.1 NormFit vs. Fine-Tuning Additional LayerNorms

Table 4 presents the experimental results for various LayerNorm fine-tuning strategies in few-shot FL with non-IID data. The results directly confirm the earlier discussions in Sections 3.1 and 3.2 as well as the theoretical analysis in Section 3.3. Specifically, as discussed in Section 3.1, fine-tuning parameters in the vision encoder is more effective than in the text encoder for adapting to distribution shifts. Section 3.2 further explained that fine-tuning only the Pre-LayerNorm enables rapid alignment of feature statistics while minimizing parameter updates, and Section 3.3 theoretically justified that fine-tuning fewer parameters leads to a significantly smaller generalization gap. Consistent with these insights, Table 4 shows that NormFit, which fine-tunes only the vision encoder's Pre-LayerNorm, achieves the best performance among all strategies.

To gain deeper insight into the effect of fine-tuning additional LayerNorms in the model, Figure 5 presents the results of progressively unfreezing the LayerNorms in both the vision and text encoders. Each encoder consists of a Pre-LayerNorm and 12 transformer layers, each containing two LayerNorms. The horizontal axis in Figure 5 represents the cumulative number of LayerNorms fine-tuned, starting from the Pre-LayerNorm of the vision encoder (NormFit), then gradually including all LayerNorms of the vision encoder, followed by the Pre-LayerNorm of the text encoder, and finally all LayerNorms of the text encoder. As more LayerNorms are fine-tuned, the performance tends to degrade, highlighting that excessive fine-tuning disrupts the pretrained alignment between the vision and text modalities in CLIP.

### 4.5.2 NormFit with Different Maximum Sample Sizes per Class in Few-Shot FL

In our FL setup, we first sample per-user class proportions from a Dirichlet distribution, then cap the resulting per-class sample counts at a predefined maximum of 16. To assess how this cap influences learning, we vary the per-class sample limit from 1 to 16 and measure the resulting accuracy. Figure 6 and Table A.3 present these results, demonstrating that the larger the cap (thus providing richer local datasets), the better NormFit performs in few-shot FL.

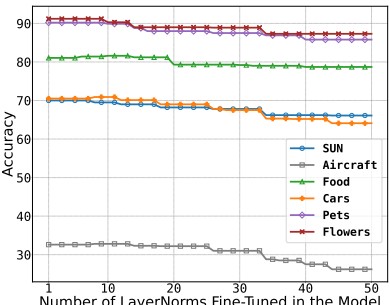
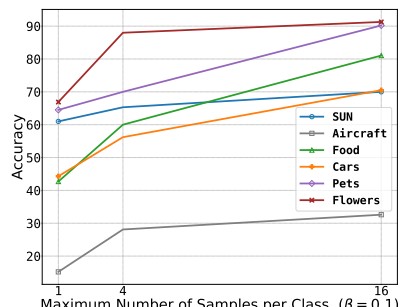

Figure 5: Fine-tuning more LayerNorms in both encoders in few-shot FL. '1' denotes fine-tuning the Pre-LayerNorm of the vision encoder (NormFit). Tuning more and deeper layers degrades the performance, underscoring the critical role of early normalization layers.

Figure 6: Impact of increasing the maximum number of samples per class on accuracy across different datasets in few-shot FL. Accuracy improves as the maximum number of samples per class increases, highlighting the benefit of more data.

Table 5: Zero-shot performance of NormFit.



(a) Trained on Food.

| Test Dataset | NormFit | All LayerNorms | Zero-shot CLIP |
|---|---|---|---|
| Food | **81.06**±0.51 | 78.80±0.49 | 76.42 |
| CIFAR-10 | **83.92**±0.36 | 76.53±0.19 | 88.73 |
| Flowers | **63.75**±0.60 | 52.49±0.02 | 66.10 |

(b) Trained on Pets.

| Test Dataset | NormFit | All LayerNorms | Zero-shot CLIP |
|---|---|---|---|
| Pets | **90.22**±0.33 | 84.24±0.09 | 85.35 |
| CIFAR-10 | **85.11**±0.42 | 74.50±0.27 | 88.73 |
| Flowers | **64.38**±0.24 | 50.43±0.10 | 66.10 |



### 4.5.3 NormFit in Zero-shot mode

Table 5(a) and (b) present the zero-shot performance of NormFit compared to fine-tuning all LayerNorms, with Food and Pets used as training datasets, respectively. The results highlight that fine-tuning all LayerNorms can degrade robustness, whereas NormFit offers a better trade-off between adaptation and generalization.

### 4.5.4 NormFit in Centralized Few-Shot Learning

While NormFit is designed for few-shot FL with non-IID data, it is also effective to the centralized setup when the few-shot training data is class-imbalanced. In Appendix E, we present the results for the centralized learning case with class-imbalanced few-shot data. Specifically, Table A.4 compares the accuracy of NormFit with that of the other few-shot fine-tuning baselines in centralized few-shot learning. Table A.5 evaluates NormFit as an add-on, and Table A.6 explores the effect of fine-tuning more LayerNorms in centralized few-shot learning.

## 5  Conclusions

We introduced NormFit, a novel framework designed to address the issue of non-IID data in few-shot FL for VLMs. By fine-tuning only the Pre-LayerNorm in the vision encoder of a VLM, NormFit effectively mitigated the bias toward majority classes while enhancing the representations of minority or underrepresented classes. NormFit demonstrated superior performance across a wide range of datasets, often achieving notable gains in accuracy compared to SOTA approaches. It substantially reduces both communication overhead and local computational costs, making it an ideal solution for bandwidth-limited FL with resource-constrained users. When integrated with traditional few-shot fine-tuning methods, NormFit further improved the performance, underscoring its versatility.

While being effective in practical non-IID FL scenarios, NormFit may be less suited for extreme domain shifts. Additionally, NormFit applies a uniform strategy across users in FL without personalizing for differences in data quality or domain. Also, in high data regime, other traditional methods may outperform NormFit in FL.

## Acknowledgments

This work was supported by the Digital Research Alliance of Canada through the Researcher Resource Grant (RRG) 5193 (2025), and by the Natural Sciences and Engineering Research Council of Canada (NSERC) through the Discovery Grant RGPIN-2025-04708.

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

**Appendix**

The Appendix includes the following items:

## A   Proof of Theorem 1

We start from the generalization gap bound given in Theorem 2 of [Mohri et al., 2019]:

$$\forall h \in \mathcal{H}, \quad \sup_{\vec{\lambda} \in \Lambda} \left[ L_{\mathcal{D}_{\vec{\lambda}}}(h) - L_{\hat{\mathcal{D}}_{\vec{\lambda}}}(h) \right] \leq 2\mathfrak{R}_{\mathbf{m}}(\mathcal{G}, \Lambda) + M\epsilon + M\sqrt{\frac{s(\Lambda\|\bar{\mathbf{m}})}{2m} \log\left(\frac{|\Lambda_\epsilon|}{\delta}\right)}, \quad \text{(A.1)}$$

where $\mathfrak{R}_{\mathbf{m}}(\mathcal{G}, \Lambda)$ denotes the weighted Rademacher complexity over the loss class $\mathcal{G}$, $\mathbf{m} = (m_1, \cdots, m_M)$ the empirical sampling distribution, $s(\Lambda\|\bar{\mathbf{m}})$ the skewness, $m = \sum_{k=1}^{M} m_k$, and $\bar{\mathbf{m}} = \frac{1}{m}\mathbf{m}$. Note that the bound can be expressed as

$$\varepsilon \leq 2\mathfrak{R}_{\mathbf{m}}(\mathcal{G}, \Lambda) + (\text{terms independent of model complexity}). \quad \text{(A.2)}$$

We now focus on bounding the dominant term $\mathfrak{R}_{\mathbf{m}}(\mathcal{G}, \Lambda)$ in terms of model capacity. By Lemma 3 in [Mohri et al., 2019], for a hypothesis class $\mathcal{H}$ with Vapnik–Chervonenkis (VC)-dimension $L$, the weighted Rademacher complexity satisfies:

$$\mathfrak{R}_{\mathbf{m}}(\mathcal{G}, \Lambda) \leq \sqrt{\frac{2L}{m} s(\Lambda\|\bar{\mathbf{m}}) \log\left(\frac{em}{L}\right)}. \quad \text{(A.3)}$$

The VC-dimension bound $L$ is upper bounded by Eq. (2) in [Bartlett et al., 2019]:

$$L \leq cpq \log(p), \quad \text{(A.4)}$$

where $c$ is a constant, $p$ is the number of trainable parameters, and $q$ is the number of layers.

We now consider two cases: fine-tuning (i) Pre-LayerNorm only and (ii) all LayerNorms including the Pre-LayerNorm in the model. First, for the case of fine-tuning the Pre-LayerNorm only, because the scale and shift of the single layer are trained ($p = 2d$ and $q = 1$), we have

$$L_{pre} \leq 2cd \log(2d), \quad \text{(A.5)}$$

and the generalization gap is upper-bounded as:

$$\varepsilon_{pre} \lesssim \sqrt{\frac{4cd\log(2d)}{m}s(\Lambda\|\bar{\mathbf{m}})\log\left(\frac{em}{2cd\log(2d)}\right)}. \tag{A.6}$$

For the second case of fine-tuning all LayerNorms including the pre-LayerNorm and those Layer-Norms in the $N$ transformer layers, we have $q = (2N + 1)$ and $p = 2d(2N + 1)$ because each transform layer has two LayerNorms. It follows that

$$L_{all} \leq 2cd(2N + 1)^2 \log(2d(2N + 1)). \tag{A.7}$$

and the generalization gap is upper-bounded as:

$$\varepsilon_{all} \lesssim \sqrt{\frac{4cd(2N + 1)^2 \log(2d(2N + 1))}{m}s(\Lambda\|\bar{\mathbf{m}})\log\left(\frac{em}{2cd(2N + 1)^2 \log(2d(2N + 1))}\right)}. \tag{A.8}$$

Their ratio becomes:

$$\frac{\varepsilon_{all}}{\varepsilon_{pre}} \lesssim \sqrt{\frac{(2N + 1)^2 \log(2d(2N + 1))}{\log(2d)} \cdot \frac{\log\left(\frac{em}{2cd(2N+1)^2 \log(2d(2N+1))}\right)}{\log\left(\frac{em}{2cd\log(2d)}\right)}}. \tag{A.9}$$

In many practical regimes (i.e., when the capacity-dependent logarithmic terms vary slowly with the model complexity), the ratio of the two logarithmic terms is upper bound by a constant. Dropping the constant, the bound is simplified to

$$\frac{\varepsilon_{all}}{\varepsilon_{pre}} \lesssim \sqrt{(2N + 1)^2 \left(1 + \frac{\log(2N + 1)}{\log(2d)}\right)}. \tag{A.10}$$

∎

## B  Proof of Theorem 2

We model each layer as a transformation of the form

$$h_l = f_l(h_{l-1}, \Theta_l) \tag{A.11}$$

where $h_0 = x$ is the input to the model and $h_l$ is the output of layer $l$. Layer-wise Lipschitz constant, $L_l$, is the supremum of the norm of its Jacobian $J$ with respect to its input, $h_{l-1}$:

$$L_l = \sup_{h_{l-1}} \|J_{f_l}(h_{l-1}, \Theta_l)\|. \tag{A.12}$$

Due to a chain of dependencies on all preceding layers, we have

$$h_{l-1} = f_{l-1}(h_{l-2}, \Theta_{l-1}) = f_{l-1}(f_{l-2}(\dots f_1(x, \Theta_1)\dots), \Theta_{l-1}). \tag{A.13}$$

Thus, the Lipschitz constant $L_l$ of layer $l$ is an implicit function of all parameters up to and including layer $l$:

$$L_l = L_l(\Theta_1, \Theta_2, \dots, \Theta_l). \tag{A.14}$$

For an arbitrary layer $k$, we take the gradient of $L_{\text{global}}$ with respect to $\Theta_k$. Applying the multivariate chain rule, we have

Table A.1: NormFit with different types of vision encoders in few-shot FL.

| Baseline | CIFAR10 | ImageNet | Caltech | OxfordPets | Cars | Flowers | Food | Aircraft | SUN | DTD | Average |
|---|---|---|---|---|---|---|---|---|---|---|---|
| NormFit (LiT) | **91.09**±0.09 | **63.12**±0.10 | **91.45**±0.07 | **91.72**±0.12 | **70.73**±0.20 | **92.26**±0.07 | **82.42**±0.11 | **33.48**±0.19 | **72.02**±0.22 | **63.16**±0.07 | **75.15**±0.12 |
| NormFit (ConvNeXt) | 88.46±0.16 | 54.62±0.20 | 87.19±0.29 | 89.54±0.21 | 71.77±0.16 | 90.08±0.18 | 79.19±0.30 | 30.55±0.41 | 70.63±0.29 | 61.39±0.10 | 72.34±0.23 |
| NormFit (ours) | 89.54±0.07 | 62.57±0.21 | 89.09±0.06 | 90.22±0.18 | 70.46±0.25 | 91.27±0.08 | 81.06±0.31 | 32.59±0.44 | 70.01±0.62 | 60.42±0.10 | 73.72±0.23 |

$$\nabla_{\Theta_k} L_{\text{global}} = \sum_{j=1}^{N} \frac{\partial L_{\text{global}}}{\partial L_j} \nabla_{\Theta_k} L_j, \tag{A.15}$$

where

$$\frac{\partial L_{\text{global}}}{\partial L_j} = \prod_{i \neq j} L_i. \tag{A.16}$$

Since the parameters $\Theta_k$ can only influence its own Lipschitz constant $L_k$ and the Lipschitz constants of subsequent layers $L_j$ $(j > k)$, the gradient term $\nabla_{\Theta_k} L_j$ is zero for all $j < k$:

$$\nabla_{\Theta_k} L_{\text{global}} = \sum_{j=k}^{N} (\nabla_{\Theta_k} L_j) \left( \prod_{i \neq j} L_i \right). \tag{A.17}$$

Separating the first term of the sum (where $j = k$) from the rest of the terms, we have

$$\nabla_{\Theta_k} L_{\text{global}} = (\nabla_{\Theta_k} L_k) \left( \prod_{i \neq k} L_i \right) + \sum_{j=k+1}^{N} (\nabla_{\Theta_k} L_j) \left( \prod_{i \neq j} L_i \right). \tag{A.18}$$

∎

## C  NormFit with Other Vision Encoders

While our experiments mainly focus on CLIP with ViT, NormFit is broadly applicable to VLMs with other vision encoders. In Table A.1, we further demonstrate its effectiveness with LiT [Zhai et al., 2022] and ConvNeXt [Liu et al., 2022], showing that NormFit consistently improves generalization.

## D  Dirichlet Distribution-Based Non-IID Partitioning

To simulate realistic non-IID data for users in FL, we adopt the widely accepted Dirichlet distribution [Yurochkin et al., 2019] to partition class-labeled data among users. This method provides a flexible and tunable way to control the degree of non-IID-ness across users. A smaller concentration parameter $\beta$ leads to higher skewness, where each user receives data from fewer classes, mimicking strong non-IID scenarios. Conversely, larger values of $\beta$ produce more balanced distributions across users. Figures A.1 and A.2 illustrate this effect by visualizing non-IID distribution across 10 users with 10 classes for two different values of $\beta = 0.1$ and $0.05$, respectively.

### D.1  NormFit with More Skewed Data Distribution in Few-Shot FL

Table A.2 presents the accuracy of NormFit and the most representative VLM few-shot methods adopted in non-IID few-shot FL with a more skewed data distribution of $\beta = 0.05$.

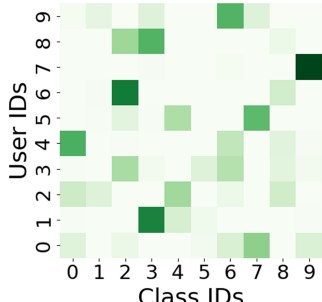

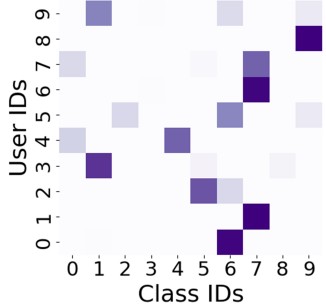

Figure A.1: Class distribution across 10 users for 10 classes with non-IID data $\beta = 0.1$.

Figure A.2: Class distribution across 10 users for 10 classes with non-IID data $\beta = 0.05$.

Table A.2: Accuracy of NormFit and the most representative VLM few-shot methods adopted in FL with more skewed data distribution ($\beta = 0.05$).

| Baseline | CIFAR10 | ImageNet | Caltech | OxfordPets | Cars | Flowers | Food | Aircraft | SUN | DTD | Average |
|----------|---------|----------|---------|------------|------|---------|------|----------|-----|-----|---------|
| CoOp | 76.27 | 47.46 | 80.39 | 71.61 | 50.11 | 80.56 | 64.18 | 19.50 | 56.40 | 54.23 | 60.07 |
| CoCoOp | 80.23 | 50.09 | 87.42 | 85.18 | 58.92 | 72.19 | 71.70 | 19.82 | 61.03 | 49.75 | 63.63 |
| TIP-Adapter | 88.37 | **61.25** | 87.64 | 85.00 | 69.74 | 88.55 | 72.17 | 30.23 | 62.94 | 56.52 | 70.24 |
| NormFit (ours) | **88.70** | 60.40 | **88.26** | **90.00** | **69.95** | **90.83** | **80.18** | **30.71** | **68.47** | **60.03** | **72.75** |

## D.2   NormFit and Different Maximum Number of Samples per Class in $\beta = 0.01$

Table A.3 shows how increasing the number of samples per class improves accuracy across three datasets (Flowers, Food, Pets) in few-shot FL with $\beta = 0.01$. The results indicate that adding more samples per class consistently boosts performance, with diminishing gains beyond 16–32 samples.

# E   NormFit in Centralized Few-Shot Learning with Class-Imbalanced Data

In this appendix, we provide the simulation results of NormFit in the centralized few-shot learning setting with class-imbalanced data (i.e., the case of a single user having class-imbalanced few-shot training data) in comparison to the existing few-shot fine-tuning methods in the class-imbalanced scenario with $\beta = 0.1$.

## E.1   NormFit as a Standalone Strategy in Centralized Few-Shot Learning with Class-Imbalanced Data

Table A.4 presents the accuracy of NormFit compared with the most representative few-shot fine-tuning methods in a centralized few-shot learning setting under class-imbalanced data conditions ($\beta = 0.1$). NormFit continues to achieve improved performance even in centralized scenarios with class imbalance, consistent with its effectiveness in non-IID few-shot FL scenarios shown previously in Table 1.

## E.2   NormFit as an Add-On to Existing Fine-Tuning Methods in Centralized Few-Shot Learning with Class-Imbalanced Data

Similar to its application in non-IID few-shot FL scenarios, NormFit can serve as an add-on to existing fine-tuning methods, further enhancing their performance in centralized few-shot learning with class-imbalanced data. Simulation results demonstrating this capability are presented in Table A.5. Specifically, fine-tuning the Pre-LayerNorm parameters of the vision encoder (NormFit) consistently improves the performance of CoOp in centralized few-shot scenarios under class imbalance.

Table A.3: Impact of increasing the maximum number of samples per class on accuracy across datasets in few-shot FL ($\beta = 0.01$).

| Dataset | 1 | 4 | 16 | 32 |
|---|---|---|---|---|
| Flowers | 66.37±0.11 | 86.39±0.41 | 87.83±0.60 | 88.11±0.50 |
| Food | 44.50±0.10 | 63.36±0.25 | 75.62±0.19 | 76.09±0.07 |
| Pets | 65.76±0.60 | 67.81±0.30 | 88.19±0.29 | 88.45±0.09 |

Table A.4: Accuracy of NormFit and most representative fine-tuning methods adopted in few-shot centralized learning.

| Baseline | CIFAR10 | ImageNet | Caltech | OxfordPets | Cars | Flowers | Food | Aircraft | SUN | DTD | Average |
|---|---|---|---|---|---|---|---|---|---|---|---|
| CoOp | 78.16 | 53.42 | 78.12 | 76.23 | 58.23 | 85.11 | 66.63 | 21.52 | 60.36 | 59.70 | 63.75 |
| CoCoOp | 82.36 | 56.30 | 85.17 | 88.26 | 63.76 | 74.50 | 75.90 | 20.32 | 64.90 | 51.59 | 64.31 |
| TIP-Adapter | 90.20 | 64.71 | 87.92 | 87.19 | 71.80 | 89.83 | 75.46 | 32.39 | 67.21 | 60.43 | 72.71 |
| NormFit (ours) | **90.76** | **65.11** | **88.50** | **90.13** | **71.92** | **91.19** | **80.93** | **34.17** | **70.24** | **61.09** | **74.40** |

Table A.5: Accuracy of NormFit as an add-on to the fine-tuning methods in few-shot centralized learning.

| Baseline | CIFAR10 | ImageNet | Caltech | OxfordPets | Cars | Flowers | Food | Aircraft | SUN | DTD | Average |
|---|---|---|---|---|---|---|---|---|---|---|---|
| CoOp | 78.16 | 53.42 | 78.12 | 76.23 | 58.23 | 85.11 | 66.63 | 21.52 | 60.36 | 59.70 | 63.95 |
| CoOp+NormFit (ours) | **91.18** | **67.77** | **89.60** | **92.27** | **73.51** | **92.43** | **81.66** | **34.04** | **69.42** | **61.29** | **75.92** |

Table A.6: NormFit vs. fine-tuning additional LayerNorms in few-shot centralized learning.

| Baseline | CIFAR10 | ImageNet | Caltech | OxfordPets | Cars | Flowers | Food | Aircraft | SUN | DTD | Average |
|---|---|---|---|---|---|---|---|---|---|---|---|
| All LayerNorms | 87.26 | 59.66 | 84.65 | 80.66 | 44.85 | 85.50 | 77.30 | 16.66 | 65.97 | 55.30 | 65.24 |
| Vision encoder LayerNorms | **91.01** | 64.51 | 88.42 | 90.03 | 70.22 | 89.23 | 80.33 | **35.92** | 68.59 | 59.32 | 70.03 |
| Text encoder LayerNorms | 86.31 | 61.98 | 87.72 | 86.24 | 66.48 | 86.49 | 78.53 | 29.70 | 66.06 | 56.90 | 66.39 |
| Text encoder Pre-LayerNorm | 89.72 | 62.25 | 88.34 | 89.44 | 71.50 | 89.01 | 73.16 | 34.47 | 67.12 | 57.78 | 67.48 |
| NormFit (ours) | 90.76 | **65.11** | **88.50** | **90.13** | **71.92** | **91.19** | **80.93** | 34.17 | **70.24** | **61.09** | **70.90** |

## E.3 Ablation study: Fine-tuning More LayerNorms in the Centralized Setting

Table A.6 presents the simulation results of fine-tuning various combinations of LayerNorm parameters in centralized few-shot settings with class-imbalanced data. Fine-tuning all LayerNorms yields overall unsatisfactory performance, notably on datasets such as Aircraft and DTD. Fine-tuning only the vision encoder LayerNorms is beneficial primarily for datasets characterized by strong visual features, such as OxfordPets. Meanwhile, fine-tuning only the text encoder LayerNorms is more effective on datasets where textual context clarifies visual concepts, such as Caltech and Food. Importantly, NormFit consistently achieves the highest accuracy across most datasets. The minor performance differences observed for CIFAR-10 and Aircraft in Table A.6 indicate that certain datasets with unique characteristics might slightly benefit from broader fine-tuning. However, these marginal improvements incur drastically higher computational costs, significantly restricting their practicality in resource-constrained scenarios. NormFit, in contrast, emphasizes minimal yet highly effective parameter updates, defining a superior balance between performance and efficiency.

