# OpenReview forum: "NormFit: A Lightweight Solution for Few-Shot Federated Learning with Non-IID Data"
_NeurIPS.cc/2025/Conference — NeurIPS 2025 spotlight_

### Official Review · Reviewer_Rw1u · 2025-06-16

**Clarity:** 3
**Significance:** 3
**Originality:** 3
**Rating:** 5
**Confidence:** 3

**Summary:**

The paper show that finetuning just the pre-LayerNorm parameters of VLMs like CLIP would reduce the communication overhead and computation cost in the problem setting of few shot federated learning, along with the improved results compared to various few shot-based methods in the same line of work. Authors show that focusing the finetuning on pre-LayerNorm parameters improve the model’s attention to object of interest compared to finetuning other LayerNorm parameters in the model. Improved results are shown across 10 different benchmarks like CIFAR10, ImageNet, Caltech etc. They also show that their proposed finetuning can also be seen as add-on to existing works.

**Questions:**

-	As the parameters to be finetuned is model dependent, and the results in Figure 1 show the CLIP ViT-B/32, how does the number of trainable parameters, communication cost and computational cost would look like for other CLIP architectures like ViT-B/16, ViT-L/14, ViT-H/14 in comparison to baselines?

-	I understand that the problem setting is focused on few-shot FL and thus the choice of 16 samples per class. Does increasing more training examples per class to 32 or 64 still keep the improvement gap of the proposed approach against baselines?

**Ethical Concerns:**

["NO or VERY MINOR ethics concerns only"]

**Final Justification:**

I find that the paper proposed a simple and effective finetuning strategy in few-shot FL that outperform existing baselines. From the rebuttal, the evidence that NormFit keeps its efficiency margin while scaling to a larger backbone is reassuring, and the diminishing-returns analysis for >16 samples/class clarifies when the method is most useful. I believe this work would be of interest to the community for its ability to reduce communication overhead and computational cost while still outperforming few shot FL baselines.

**Limitations:**

Yes

**Quality:**

3

**Strengths And Weaknesses:**

Strengths:

-	It is a simple but effective finetuning strategy utilizing existing parameters in few-shot FL that is shown to outperform existing baselines which introduce additional set of parameters.

-	This choice of pre-LayerNorm finetuning shown to greatly reduce the communication overhead and computational cost at the user side compared to previous works. To the best of my knowledge, this is the first work to show that adapting a small set of parameters of a layer would result in improvements and reduce computational cost that turn out to be beneficial in the few shot federated learning setting.

-	Authors shown the effect of finetuning only pre-LayerNorm result in semantically aligned attention on object of interest compared to finetuning other LayerNorm layers. I appreciate this analysis as it sheds some understanding on the benefit of finetuning this specific layer.

-	Experiments are conducted on 10 different benchmarks, compared against various baselines like FedCLIP, FedTPG, FedDAT and FLoRA, and demonstrated improvements over them.

-	Paper is also easy to read and follow through the storyline.

Weakness:

-	I don’t find major weaknesses in the paper, as the paper try to understand the effect of pre-LayerNorm through Grad-CAM activations and makes a discussion on the benefits of this specific choice, compare against SOTA methods across multiple datasets, and also conduct ablation study on finetuning additional layer norms. However, I have few questions (please refer to Questions below).

---

> ### Author Rebuttal · Authors · 2025-07-30
>
> **Thank you very much for your thoughtful and constructive feedback.**
>
> **[Question 1]**
>
> We have conducted additional experiments using a larger CLIP architecture, ViT-B/16, selected for rebuttal analysis due to evaluation feasibility within the review window.
>
> The table below summarizes the number of trainable parameters, communication cost, and computation cost for NormFit compared to the two best-performing baselines: TIP-Adapter and FLoRA. Given the limited time available during the rebuttal period, we focused on these two baselines because they are the most competitive and meaningful baselines. Specifically, FLoRA is currently the best performing (SOTA) method and also the most lightweight among all baseline methods and TIP-Adapter is the second-best performing method and remains relatively lightweight.
>
> As shown in the table, NormFit consistently uses dramatically fewer trainable parameters and incurs far lower communication and computation costs.
>
> In the final version of the paper, complete results for other baselines and additional results for ViT-L/14 and ViT-H/14 will be included in an appendix.
>
> **Table: Comparison of communication, computation, and parameter overhead in few-shot FL settings with CLIP with ViT-B/16.**
>
> | Method               | Trainable Params (K) | Comm. Cost (KB) | Comp. Cost (GFLOPs) |
> |----------------------|----------------------|------------------|----------------------|
> | TIP-Adapter        | 7.9 × 10²            | 2.4 × 10²        | 1.2 × 10⁶            |
> | FLoRA                | 2.5 × 10¹            | 9.6 × 10¹        | 1.1 × 10⁶            |
> | **NormFit (ours)**   | **1.5**              | **4.1**          | **4.9 × 10²**        |
>
> **[Question 2]**
>
> We have conducted additional experiments on three datasets for the case of 32 samples/class. The results indicate that, with NormFit, increasing the number of samples per class from 16 to 32 continues to improve accuracy. However, the gains beyond 16 samples are relatively modest, suggesting diminishing returns. In the final version of the paper, additional results for 64 samples/class will be included.
>
> **Table: NormFit with increasing number of samples per class on accuracy in few-shot FL ($\beta$ = 0.01)**
>
> | Dataset  |   1 sample/class   |  4 samples/class  | 16 samples/class  | 32 samples/class  |
> |----------|--------------------|-------------------|-------------------|-------------------|
> | Flowers  |   66.37 ± 0.11     |   86.39 ± 0.41    |   87.83 ± 0.60    |   88.11 ± 0.50    |
> | Food     |   44.50 ± 0.10     |   63.36 ± 0.25    |   75.62 ± 0.19    |   76.09 ± 0.07    |
> | Pets     |   65.76 ± 0.60     |   67.81 ± 0.30    |   88.19 ± 0.29    |   88.45 ± 0.09    |

---

> > ### Comment · Reviewer_Rw1u · 2025-08-07
> >
> > The additional experiments with ViT-B/16 and the 32-samples-per-class setting strengthen the paper and answer my main technical questions. The evidence that NormFit keeps its efficiency margin while scaling to a larger backbone is reassuring, and the diminishing-returns analysis for >16 samples/class clarifies when the method is most useful. Seeing TIP-Adapter and FLoRA in the same 32-shot and 64-shot setting would quantify the gap as data grows, and those numbers will help to understand when these baselines would catch up to NormFit. My evaluation remains positive. The rebuttal addressed the critical points. I am inclined to keep my original score and suggest the authors to include these additional results in their final verison.

---

> > > ### Author Response · Authors · 2025-08-07
> > >
> > > Thank you very much for the thoughtful feedback. We're glad the additional experiments helped address your concerns. We will surely include the results for TIP-Adapter and FLoRA under the same 32- and 64-shot settings in the final version.

---

### Official Review · Reviewer_Qst8 · 2025-06-24

**Clarity:** 3
**Significance:** 3
**Originality:** 3
**Rating:** 5
**Confidence:** 4

**Summary:**

This paper introduces NormFit, a novel and lightweight fine-tuning method for adapting Vision-Language Models (VLMs) like CLIP in the context of few-shot Federated Learning (FL) with non-IID data. The core problem addressed is the trade-off between model performance, communication overhead, and local computation costs, which plagues existing methods. NormFit's key idea is its simplicity and efficiency: it selectively fine-tunes only the Pre-LayerNorm parameters of the vision encoder, while keeping the rest of the VLM frozen.

**Questions:**

1. In Table 4, does the results of "NormFit as an Add-On" applying Tip-Adapter require extra hyper-parameter searching?
2. Does a larger number of samples per class (>16) achieve higher accuracy in Figure 6?
3. Can NormFit be transferred to CNN-based encoders?

**Ethical Concerns:**

["NO or VERY MINOR ethics concerns only"]

**Final Justification:**

The author's rebuttal partly resolved my doubts, and I decided to maintain the previous rating unchanged.

**Limitations:**

Please refer to the questions section.

**Quality:**

3

**Strengths And Weaknesses:**

Strengths
1. The proposed method, NormFit, is elegant in its simplicity. While parameter-efficient fine-tuning (PEFT) is a well-established field, the specific choice to tune only the Pre-LayerNorm of the vision encoder is a novel and non-obvious insight. This simplicity makes it easy to implement without any changes to model architecture or complex algorithms.
2. The paper presents compelling evidence that NormFit resolves the critical trade-off between accuracy and efficiency in few-shot FL. The results in Figure 1 and Tables 1 & 2 are particularly strong, showing that NormFit simultaneously achieves the highest accuracy and the lowest communication/computation costs among all compared methods.

Weaknesses
1. The experiments are primarily focused on the CLIP model with a Vision Transformer (ViT) or Locked-image Text tuning (LiT) backbone. While these are representative, the paper's claims of general applicability would be stronger if the principle was tested on other VLM architectures or even different model families (e.g., CNN-based vision encoders) to demonstrate broader robustness.
2. The generalization gap analysis is a significant strength, but it primarily explains why tuning fewer parameters is better. It does not fully provide a theoretical reason for the specific effectiveness of the Pre-LayerNorm layer over other layers, beyond the intuition that it is at an "early stage".

---

> ### Author Rebuttal · Authors · 2025-07-30
>
> **Thank you very much for your thoughtful and constructive feedback.**
>
> **[Weakness 1, Question 3]**
>
> NormFit is potentially applicable to any models having normalization parameters, because the core idea of NormFit is to fine-tune a minimal set of normalization parameters at an early-stage.
>
> Regarding applicability to CNNs, recent advancements have made NormFit directly applicable to modern, high-performance, SOTA convolutional architectures like ConvNeXt. Designed to compete with Transformers, ConvNeXt has explicitly adopted Layer Normalization. Since NormFit targets fine-tuning early layer normalization parameters, it can be seamlessly applied to a ConvNeXt-based vision encoder without modification.
>
> Conceptually, the principle of NormFit also extends to traditional CNNs like ResNet. A direct application would be to fine-tune only the affine parameters (gamma and beta) of early Batch Normalization (BN) layers, offering a principled and lightweight alternative to ad-hoc fine-tuning practices.
>
> During the rebuttal period, we have identified suitable open-source, pre-trained VLMs based on ConvNeXt (e.g., OLA-VLM) and ResNet to specifically test and apply NormFit in our forthcoming experiments. Unfortunately, due to the time constraint, we were unable to run and include these experiments in this rebuttal. However, all authors are committed to conducting these experiments on VLMs with ConvNeXt and ResNet encoders, and the results will be surely included in the final version of the paper.
>
> **[Weakness 2]**
>
> We fully agree that Theorem 1 offers only theoretical support for the benefit of fine-tuning a single LayerNorm over fine-tuning multiple or all LayerNorms. Complementing Theorem 1, we now introduce additional new and significant theoretical result, Theorem 2, to analytically show why the optimal choice must be the first one (i.e., the Pre-LayerNorm), instead of one located deeper in the network. This new result highlights the unique influence of the first LayerNorm within the network. To formalize this locational advantage, we leverage Lipschitz constants, which are an essential tool in deep neural network analysis.
>
> A function $f: \mathbb{R}^n \to \mathbb{R}^m$ is defined as Lipschitz continuous if there exists a non-negative constant $L$ such that for any two inputs $x_1, x_2$, the inequality $\|\|f(x_1) - f(x_2)\|\| \le L \|\|x_1 - x_2\|\|$ holds. The smallest such $L$  is *the* Lipschitz constant of $f$.
> In the following theorem, we provide the gradient of the global Lipschitz bound with respect to the parameters of any arbitrary LayerNorm (including the Pre-LayerNorm as a special case).
>
> **Theorem 2 (Influence of fine-tuning a LayerNorm)**
> Let $f_{\Theta}$ be a ViT of $N$ layers, $f = f_N \circ \dots \circ f_1$, where  $\Theta$ represents the set of trainable parameters of all LayerNorms in the ViT. All other parts including attention blocks and MLPs are frozen.  Let $L_{global}(\Theta) = \prod_{l=1}^{N} L_l(\Theta)$ be an upper bound on its global Lipschitz constant, where $L_k(\Theta)$ is the Lipschitz constant of layer $k$. Let $\Theta_k$ denote the set of trainable parameters of the $k$-th LayerNorm (in layer $k$). The gradient of the global Lipschitz bound with respect to $\Theta_k$ is given by:
> $$\nabla_{\Theta_k} L_{global} = DIT_k + CIT_k$$
> where the Direct Influence Term, $DIT_k$, is given by
> $$DIT_k= \left( \nabla_{\Theta_k} L_k \right) A_k$$
> with  amplification factor
> $$A_k=\left( \prod_{i =1}^{k-1} L_i \right) \left( \prod_{i = k+1}^N L_i \right)$$
> and the Cascading Influence Term, $CIT_k$, is given by
> $$CIT_k=\sum_{j=k+1}^{N} \left( \nabla_{\Theta_k} L_j \right) \left( \prod_{i \ne j} L_i \right)$$
>
>
> **Proof:**
> Each layer is represented by transformation $h_l = f_l(h_{l-1}, \Theta_l)$,
> where $h_0 = x$ is the input to the model and $h_l$ is the output of layer $l$. Layer-wise Lipschitz constant, $L_l$, is the supremum of the norm of its Jacobian $J$ with respect to its input, $h_{l-1}$:
> $$L_l = \sup_{h_{l-1}} \|\| J_{f_l}(h_{l-1}, \Theta_l) \|\|$$
> Due to a chain of dependencies on all preceding layers, we have
> $$h_{l-1} = f_{l-1}(h_{l-2}, \Theta_{l-1}) = f_{l-1}(f_{l-2}(\dots f_1(x, \Theta_1)\dots), \Theta_{l-1})$$
> Thus, the Lipschitz constant $L_l$ of layer $l$ is an implicit function of all parameters up to and including layer $l$:
> $$L_l = L_l(\Theta_1, \Theta_2, \dots, \Theta_l)$$
>
> For an arbitrary layer $k$, we take the gradient of $L_{global}$ with respect to $\Theta_k$. Applying the multivariate chain rule, we have
> $$\nabla_{\Theta_k} L_{global} = \sum_{j=1}^{N} \frac{\partial L_{global}}{\partial L_j} \nabla_{\Theta_k} L_j$$
> where
> $$\frac{\partial L_{global}}{\partial L_j} = \prod_{i \ne j} L_i$$
> Since the parameters $\Theta_k$ can only influence its own Lipschitz constant $L_k$ and the Lipschitz constants of subsequent layers $L_j$ ($j > k$), the gradient term $\nabla_{\Theta_k} L_j$ is zero for all $j < k$:
>     $$\nabla_{\Theta_k} L_{global} = \sum_{j=k}^{N} (\nabla_{\Theta_k} L_j) \left( \prod_{i \ne j} L_i \right)$$
> Separating the first term of the sum (where $j=k$) from the rest of the terms, we have
>     $$\nabla_{\Theta_k} L_{global} = \left( \nabla_{\Theta_k} L_k \right) \left( \prod_{i \ne k} L_i \right) + \sum_{j=k+1}^{N} \left( \nabla_{\Theta_k} L_j \right) \left( \prod_{i \ne j} L_i \right)$$
>
> $\blacksquare$
>
> Theorem 2 clearly reveals the unique locational influence of Pre-LayerNorm. First, in the Direct Influence Term, the amplification factor has the property of $A_1 > A_k$ for $k>1$, which means that parameters of the first LayerNorm possess a uniquely amplified influence on the global Lipschitz bound, positioning them at a point of maximum leverage for model regularization. We now show why $A_1 > A_k$ holds. The amplification factor  is  $A_1 = \prod_{j=2}^{N} L_j$ for $k=1$, while it is $A_k = \prod_{j\neq k} L_j = (L_1/ L_k) A_1$ for $k > 1$. Clearly,  $A_1 > A_k$ holds if and only if $L_1 < L_k$. Pre-LayerNorm at $k = 1$ receives input directly from the patch embeddings, which have a well-controlled, low-variance distribution. In contrast, a LayerNorm at any deeper layer $k > 1$ receives hidden states that have been progressively amplified by the preceding $k -1$ transformer blocks. These hidden states exhibit higher variance due to the accumulation of nonlinear transformations and residual connections. To handle this increased variance, deeper layers must be more expansive, leading to larger effective Lipschitz constants; that is, $L_1 < L_k$.
>
> Second, the Cascading Influence Term is also different. For the Pre-LayeNorm ($k=1$), it is a sum of terms describing how changes in $\Theta_1$ propagate to affect the Lipschitz constants of all $(N-1)$ subsequent layers, $2, \dots, N$. In contrast, for any deeper layer $k > 1$, it is a sum of terms describing how changes in $\Theta_k$ propagate to affect the Lipschitz constants of only the $(N-k)$ subsequent layers, ${k+1}, \dots, N$.
>
>
> Intuitively, changes to the Pre-LayerNorm parameters $(k=1)$ have a direct impact that is amplified by the entire depth of the network, alongside a cascading effect that ripples through every subsequent layer. This provides a powerful, global control mechanism. In contrast, changes to deeper LayerNorm parameters $(k>1)$ have a direct impact that is amplified by a structurally different, and smaller, product of constants. Also, the cascading effect is more localized, influencing only the remaining deeper layers.
>
> **[Question 1]**
>
> For the question “*In Table 3, does the results of "NormFit as an Add-On" applying Tip-Adapter require extra hyper-parameter searching?*”, the answer is no. The results for “NormFit as an Add-On applied to Tip-Adapter” were obtained using Tip-Adapter’s existing training setup, without any extra hyper-parameter search or special adjustments. This is because NormFit does not introduce new hyper-parameters or external modules of its own: NormFit simply fine-tunes the Pre-LayerNorm parameters already present in the model. In other words, NormFit operates within the original configuration of Tip-Adapter, so no separate tuning was necessary to integrate the two.
>
> As the paper emphasizes, the performance gains obtained by adding NormFit were achieved without requiring any algorithmic changes, architectural modifications, or additional parameters, underscoring that we did not perform any extra hyper-parameter sweeping for the combined method.
>
> **[Question 2]**
>
> We have conducted additional experiments on three datasets for the case of 32 samples/class. The results indicate that, with NormFit, increasing the number of samples per class from 16 to 32 continues to improve accuracy. However, the gains beyond 16 samples are relatively modest, suggesting diminishing returns. In the final version of the paper, additional results for 64 samples/class will be included.
>
> **Table: NormFit with increasing number of samples per class on accuracy in few-shot FL ($\beta$ = 0.01)**
>
> | Dataset  |   1 sample/class   |  4 samples/class  | 16 samples/class  | 32 samples/class  |
> |----------|--------------------|-------------------|-------------------|-------------------|
> | Flowers  |   66.37 ± 0.11     |   86.39 ± 0.41    |   87.83 ± 0.60    |   88.11 ± 0.50    |
> | Food     |   44.50 ± 0.10     |   63.36 ± 0.25    |   75.62 ± 0.19    |   76.09 ± 0.07    |
> | Pets     |   65.76 ± 0.60     |   67.81 ± 0.30    |   88.19 ± 0.29    |   88.45 ± 0.09    |

---

> > ### Comment · Reviewer_Qst8 · 2025-08-01
> >
> > I am very grateful to the author for his efforts in the rebuttal process. The author's rebuttal partly resolved my doubts, and I decided to maintain the previous rating unchanged.

---

> > > ### Author Response · Authors · 2025-08-07
> > >
> > > Thank you for your response and for considering our rebuttal. We really appreciate your engagement and feedback throughout the review process.

---

### Official Review · Reviewer_s3pM · 2025-07-02

**Clarity:** 3
**Significance:** 3
**Originality:** 3
**Rating:** 4
**Confidence:** 4

**Summary:**

This paper proposes NormFit, a lightweight fine-tuning strategy for the CLIP model in few-shot Federated Learning settings with non-IID data. The core idea is to selectively fine-tune only the Pre-LayerNorm parameters of the vision encoder, while keeping other parameters frozen. This approach significantly reduces communication overhead and local computational demands, while simultaneously achieving superior accuracy, particularly for underrepresented classes in non-IID scenarios. A theoretical justification for a smaller generalization gap is provided, and empirical results on various datasets demonstrate NormFit's effectiveness both as a standalone method and as an add-on to existing fine-tuning techniques.

**Questions:**

As pointed out in the weaknesses, I have the following main concerns:

1. While fine-tuning only the Pre-LayerNorm parameters demonstrates effectiveness, the approach is relatively simple and lacks strong connections to the federated learning scenario. The paper would benefit from a more detailed discussion on the research motivation, as well as the relationship between the proposed method and key aspects of federated learning such as privacy preservation and data security.
2. Given the stochastic nature of data distributions under non-IID settings, repeating experiments and reporting error bars or confidence intervals would significantly enhance the credibility of the results.
3.  The comparison with CoOp and CoCoOp is insufficient, as both methods are outdated and focus solely on fine-tuning the text encoder of CLIP. This differs significantly from NormFit. Furthermore, comparisons with non-VLM-based federated learning approaches are also necessary to better contextualize the contribution of the proposed method.
4. The authors claim that fine-tuning only the Pre-LayerNorm helps preserve prior knowledge; however, this claim is not supported by experimental evidence. Including an evaluation of the model’s zero-shot performance after fine-tuning would help substantiate this argument and improve the quality of the paper.

Additionally, the reviewer has a question regarding**Figure B.2**. It appears that, for class 3, no samples were assigned to any of the users. Is this a valid outcome under the considered data partitioning strategy? Alternatively, could it be due to low color contrast in the figure? Clarification on this point would be helpful.

**Ethical Concerns:**

["NO or VERY MINOR ethics concerns only"]

**Final Justification:**

The authors have addressed the concerns in my initial review. Therefore, I revised my score upwards accordingly.

**Limitations:**

The authors discuss the limitations of the paper to a certain extent, such as the inability to adapt to excessive domain shift and the lack of personalization ability, but do not mention the solution ideas and future research content. The reviewers suggest further discussion on how NormFit could be extended or combined with personalization techniques within FL.

**Quality:**

2

**Strengths And Weaknesses:**

*Strengths*:

1. The inclusion of a theoretical analysis on the generalization gap provides a valuable mathematical foundation for why NormFit performs robustly with minimal parameter updates. This strengthens the paper's claims beyond empirical observations.
2. NormFit achieves improvements without requiring algorithmic modifications, architectural changes, or additional external parameters, enabling seamless integration with existing FL pipelines.
3. Extensive experiments across diverse datasets and comparisons with state-of-the-art methods (e.g., FedCLIP, FLoRA) demonstrate consistent improvements in accuracy, communication efficiency, and computational cost.

*Weaknesses*:

1. The novelty of the work is incremental. While updating only the Pre-LayerNorm parameters of the vision encoder demonstrates effectiveness, the approach is relatively straightforward and does not constitute a fundamentally new algorithm. Similar strategies have already been applied in \[1].
2. The proposed method has limited relevance to key concerns in federated learning such as privacy and security. The focus is solely on communication efficiency in distributed learning frameworks, which weakens the motivation and scope of the study.
3. The experimental results lack statistical rigor. Only single-run results or average accuracies are reported, without error bars or confidence intervals. Given the stochasticity of data distributions in non-IID settings, this raises concerns regarding the robustness and generalizability of the observed performance improvements.
4. The baseline methods used for comparison—CoOp and CoCoOp—are outdated, and both involve fine-tuning only the text encoder of CLIP, which differs significantly from NormFit’s strategy. A more appropriate comparison would be with \[2]. Moreover, comparisons with non-VLM-based federated learning approaches are also needed to better contextualize the contributions.
5. The authors claim that fine-tuning only the Pre-LayerNorm helps preserve prior knowledge; however, this assertion lacks further analysis or theoretical justification.

\[1] Tent: Fully Test-Time Adaptation by Entropy Minimization (ICLR 2021)

\[2] MaPLe: Multi-modal Prompt Learning (CVPR2023)

---

> ### Author Rebuttal · Authors · 2025-07-30
>
> **Thank you very much for your thoughtful and constructive feedback.**
>
> **[Weakness 1]**
>
> We respectfully emphasize that the novelty of NormFit lies in its *innovative* identification of Pre-LayerNorm as the optimal fine-tuning target—an approach that is *fundamentally* distinct from prior tuning strategies. Furthermore, we now introduce additional new and significant theoretical result (Theorem 2) to analytically show why the Pre-LayerNorm is the optimal target for fine-tuning. The result complements Theorem 1. Please see our response to Reviewer **Qst8**.
>
> Regarding TENT [1], comparison with TENT actually highlights the novelty of NormFit. While it is true that both methods touch normalization-related parameters, NormFit exactly *rejects* TENT’s strategy of tuning ALL normalization layers. Instead, NormFit shows that only Pre-LayerNorm of ViT should be fine-tuned in few-shot FL. Applying TENT-style full fine-tuning (whether ViT, the text encoder, or both) notably worsens accuracy compared NormFit (Table 4 and Figure 5). Moreover, such broad tuning increases communication and computing burden by orders of magnitude — clearly undesired in FL. Theoretically, Theorem 1 provides a formal justification that TENT’s style incurs a significantly larger generalization gap (up to 30×) compared to NormFit.
>
> The root cause of this divergence lies in the fundamental differences of the problem formulation, system setting, learning objective, and operating constraints. TENT is designed for *centralized, unsupervised*, and *test-time adaptation* in *inference*. To compensate for the absence of supervision, TENT updates ALL normalization layers to minimize prediction entropy on-the-fly. In contrast, NormFit operates in a *decentralized (federated), supervised, few-shot*, and *training* regime. In this context, updating all normalization layers introduces a risk of overfitting and violates efficiency requirements of FL. NormFit addresses all these issues.
>
> **[Weakness 2, Question 1]**
>
> We respectfully argue that “being simple” is a critical strength (not weakness), especially for real-world FL deployments. Simple and lightweight methods are far more likely to be adopted—especially at scale—than those requiring complex changes or coordination. In this spirit, NormFit’s design is *intentionally* minimal yet highly effective.
>
> NormFit resolves three *key* concerns of *FL* (not just communication efficiency): (1) achieving SOTA accuracy, which is unquestionably crucial; (2) reducing communication overhead, which is essential for scalability in practical FL deployments; and (3) minimizing local computation, which is critical for resource-constrained users (e.g., edge devices). Importantly, NormFit achieves the highest efficiency while outperforming current SOTA methods in accuracy.
>
> Regarding privacy and security, NormFit is orthogonal to—and fully compatible with—any existing or emerging security and privacy mechanisms such as differential privacy, secure aggregation, and homomorphic encryption (HE). NormFit’s lightweight design makes these mechanisms implementable in real-world FL settings. For example, HE is notoriously resource-intensive and often impractical for complex FL setups. The minimalism of NormFit makes HE far more feasible in practice.
>
> Importantly, NormFit’s extreme efficiency brings significant benefits to privacy and security. Its ultra-low communication footprint reduces the risk of gradient inversion attacks due to its severe information bottleneck. Reconstructing high-dimensional inputs from a very low-dimensional update vector (~ 1.5K parameters) can be a difficult inverse problem, making high-fidelity recovery challenging for an adversary. Crucially, the parameters that NormFit updates—the scale and shifts of the Pre-LayerNorm—encode low-level statistical information rather than high-level semantic content. These parameters capture very coarse-grained statistics at the *earliest* stage, making it more difficult for an adversary to extract semantically meaningful inputs from their gradients. In contrast, methods that tune adapters or prompts modify high-level feature representations that are richer in semantic detail, and thus more susceptible to inversion.
>
> Furthermore, NormFit’s design provides architectural robustness against model poisoning attacks. Such attacks typically rely on manipulating high-capacity components of the model to introduce backdoors or malicious behaviors. NormFit’s extremely constrained update space restricts the expressive capacity available to a malicious client. Attackers are, by design, constrained from modifying core components such as attention blocks or MLP layers that govern the model’s semantic reasoning and classification behavior.
>
> **[Weakness 3, Question 2]**
>
> All simulation results in the paper were obtained by running each experiment three times and reporting the average. In this rebuttal, we include the standard error of the mean (SEM) to quantify variability. Due to the rebuttal time constraints, we prioritized computing SEMs for the two strongest competing baselines. NormFit consistently outperforms these methods, with margins that mostly exceed their SEMs. For the results where SEM could not be computed within time, we use "x.xx" as a placeholder. In the final version, we will report SEMs for all results.
>
> **Table: Accuracy of NormFit and VLM few-shot methods in FL (selected datasets)**
>
> | Baseline       | Caltech        | Flowers         | Food           | Aircraft        | DTD            | SUN             |
> |----------------|----------------|------------------|----------------|------------------|----------------|------------------|
> | TIP-Adapter    | 88.11 ± 0.94   | 89.01 ± 0.66     | 72.61 ± 0.60   | 30.61 ± 0.32     | 57.25 ± 0.50   | 63.84 ± x.xx     |
> | FLoRA          | 87.93 ± 0.27   | 90.76 ± 0.09     | 79.35 ± 0.16   | 32.22 ± 0.25     | 59.09 ± 0.14   | 67.16 ± 0.12     |
> | **NormFit (ours)** | **89.09 ± 0.06** | **91.27 ± 0.08** | **81.06 ± 0.31** | **32.59 ± 0.44** | **60.42 ± 0.10** | **70.01 ± 0.62** |
>
> **[Weakness 4, Question 3]**
>
> We would like to clarify that our NormFit consistently and substantially outperforms FedTPG (Table 2 of our paper), which was published at ICLR 2024 and cited explicitly as [Qiu et al., 2024]. In Tables 1–3 of [Qiu et al., 2024], MaPLe is adapted to FL (referred to as FedMaple) and directly compared to FedTPG. The results show that FedMaple substantially underperforms FedTPG, and even underperforms FedCoOp. The core issue is that federated averaging breaks the V–L coupling that MaPLe relies on. While V–L coupling is effective in centralized learning, it becomes problematic in FL. For this reason, we did not include FedMaple as a baseline in our study. By contrast, we re-implemented CoOp and CoCoOp in our FL setup, as they demonstrated stronger performance than MaPLe in FL and could be more meaningful baselines.
>
> The current SOTA method in FL is FLoRA, and it consistently outperforms FedTPG (Table 2 of our paper). Crucially, NormFit, consistently outperforms FLoRA (Table 2 of our paper), establishing a new SOTA. Furthermore, FedTPG itself outperforms FedCoOp, which in turn outperforms FedMaple [Qiu et al., 2024], resulting in:
>
> **NormFit > FLoRA > FedTPG > FedCoOp > FedMaple**
>
> Applying MaPLe in FL also raises critical security and privacy risks. The V–L coupling produces information-rich “super-gradients,” which are more vulnerable to gradient inversion attacks. This coupling also introduces potential vectors for cross-modal leakage, where private data from one modality could be reconstructed using updates from the other.
>
> Furthermore, MaPLe introduces 3.55M additional parameters, significantly increasing communication and computation overheads—detrimental for FL. The added complexity also elevates the risk of membership inference attacks.
>
> Regarding non-VLM FL methods, a direct comparison would be uninformative due to fundamental differences in problem formulation and assumptions. In the traditional Non-VLM FL, the question is: How can we collaboratively train a relatively small, task-specific model (e.g., a ResNet) from scratch on decentralized data? The core challenges revolve around achieving model convergence under statistical heterogeneity. By contrast, in VLM-based FL (NormFit's Domain), the question is: How can we efficiently adapt a pre-trained foundation model to downstream tasks using decentralized, few-shot data? The challenges here are minimizing the crippling communication/computation costs and adapting the model's vast prior knowledge effectively without catastrophic forgetting. These two paradigms are not direct competitors.
>
> **[Weakness 5, Question 4]**
>
> We conducted the requested zero-shot evaluations after fine-tuning. First, the results demonstrate that NormFit is an effective adaptation method, outperforming baseline approaches on the held-out splits of the training datasets. Importantly, NormFit provides strong zero-shot performance on unseen datasets, supporting that catastrophic forgetting is effectively mitigated.
>
> **Table 2: Zero-Shot Performance**
>
> | Training Dataset | Testing Dataset | NormFit | All LayerNorms | Zero-shot CLIP |
> | :--- | :--- | :---: | :---: | :---: |
> | **Food** | Food | $81.06 \pm 0.51$ | $78.80 \pm 0.49$ | $76.42$ |
> | | CIFAR-10 | $83.92 \pm 0.36$ | $76.53 \pm 0.19$ | $88.73$ |
> | | Flowers | $63.75 \pm 0.60$ | $52.49 \pm 0.02$ | $66.10$ |
> | — | — | — | — | — |
> | **Pets** | Pets | $90.22 \pm 0.33$ | $84.24 \pm 0.09$ | $85.35$ |
> | | CIFAR-10 | $85.11 \pm 0.42$ | $74.50 \pm 0.27$ | $88.73$ |
> | | Flowers | $64.38 \pm 0.24$ | $50.43 \pm 0.10$ | $66.10$ |
>
> **[Question for Figure B.2]**
>
> Yes, it is due to lower color contrast in the figure. We will improve this in the final version.
>
> **[Limitations]**
>
> A promising direction is to combine NormFit with client-specific adaptation modules (e.g., lightweight task-specific heads) trained locally to capture user-specific nuances.

---

> > ### Comment · Reviewer_s3pM · 2025-08-04
> > **Official Comment by Reviewer s3pM**
> >
> > Thank you for your thorough and thoughtful rebuttal. Your detailed responses have addressed the concerns in my initial review. I have revised my score upwards accordingly.

---

> > > ### Author Response · Authors · 2025-08-07
> > >
> > > Thank you for your kind words and for taking the time to reconsider your evaluation. We're glad our responses helped clarify the concerns, and we truly appreciate your updated assessment.

---

### Note · Authors · 2025-08-11

Dear Area Chairs,

NormFit establishes new state-of-the-art (SOTA) performance in accuracy for few-shot federated learning (FL) with non-IID data by tuning only the *first* LayerNorm in the vision encoder. Importantly, the SOTA accuracy is achieved while reducing communication and computation costs by orders of magnitude, without altering architecture or adding parameters. The design is intentionally minimal, which is a core strength for real-world FL where deployment simplicity often drives adoption.

During rebuttal, we addressed all reviewer concerns with substantial new content:

•	**New mathematical results:** Theorem 2 provides a rigorous theoretical justification for the optimality of tuning only the first LayerNorm, complementing Theorem 1’s generalization gap analysis and explaining NormFit’s consistent superiority.

•	**New experimental results:** Scaling to a larger CLIP (ViT-B/16) confirmed efficiency advantages persist; zero-shot evaluation demonstrated strong prior knowledge retention; multi-run results with error bars confirmed statistical robustness; and evaluation on higher-shot settings (32 samples/class) showed continuing improvement with modest diminishing returns.

These additions directly resolved reviewer questions. Reviewer **s3pM** explicitly confirmed their concerns were addressed and raised their score. Reviewers **Qst8** and **Rw1u** both initially gave high scores (5: Accept) and upheld them post-rebuttal. This clear consensus reflects broad agreement on NormFit’s novelty, rigor, and practical value.

NormFit’s plug-and-play minimalism integrates seamlessly into existing FL pipelines and is fully compatible with privacy/security techniques, while inherently limiting semantic leakage by transmitting only minimal parameter updates. It simultaneously advances accuracy, scalability, and deployability, which breaks the traditional trade-off in few-shot FL.

We thank the ACs and reviewers for their thoughtful engagement. We trust that the demonstrated novelty, theoretical foundation, and practical impact of NormFit are clear, and we appreciate your consideration in the final evaluation.

Sincerely,

Authors

---

### Decision · Program_Chairs · 2025-09-17

**Decision:**

Accept (spotlight)

**Comment:**

The paper proposes NormFit, a minimal PEFT strategy that tunes only the Pre-LayerNorm of the vision encoder in VLMs for few-shot FL with non-IID data. Reviewers find the work technically solid with clear empirical gains and strong efficiency: it matches/exceeds SOTA accuracy while dramatically reducing communication and local compute. Post-rebuttal, two reviews remain at Accept (5) and the third explicitly raised its score after concerns were addressed. No significant ethics issues were noted. Thus, the AC agrees with the reviewers to accept the submission.